# July 14th 2016 Nice Terrorist Attack Court Trial: A Protocol on Sleep Quality and Somatic Symptoms as Markers of Risk for Traumatic Reactivation in Adolescents Exposed to This Attack

**DOI:** 10.3390/healthcare11222953

**Published:** 2023-11-12

**Authors:** Radia Zeghari, Morgane Gindt, Jokthan Guivarch, Philippe Auby, Philippe Robert, Julie Rolling, Carmen Schröder, Petri Valo, Florence Askenazy, Arnaud Fernandez

**Affiliations:** 1Nice Pediatric Psychotrauma Center (NPPC), Child and Adolescent Psychiatry Department, Hôpitaux Pédiatriques Universitaires Lenval, 06200 Nice, France; 2CoBTeK (Cognition-Behaviour-Technology) Lab, Université Cote d’Azur, 06000 Nice, France; 3Department of Child Psychiatry, APHM, 13009 Marseille, France; jokthan.guivarch@ap-hm.fr; 4CANOP Institut de Neurosciences de la Timone, UMR 7289, CNRS, Aix Marseille University, 13005 Marseille, France; 5Faculty of Medicine, Aix-Marseille University, 13005 Marseille, France; 6Regional Center for Psychotraumatism Great East, Strasbourg University Hospital, 67000 Strasbourg, France; 7Department of Child and Adolescent Psychiatry, Strasbourg University Hospitals, 67000 Strasbourg, France; 8CNRS UPR3212-Research Team “Light, Circadian Rhythms, Sleep Homeostasis and Neuropsychiatry”, Institute of Cellular and Integrative Neurosciences, 67000 Strasbourg, France; 9Excellence Centre for Autism and Neurodevelopmental Disorders STRAS&ND, 67091 Strasbourg, France; 10Sleep Disorders Centre & International Research Centre for ChronoSomnology (Circsom), University Hospitals Strasbourg, 67091 Strasbourg, France; 11Expert Centre for High-Functioning Autism, Fondation FondaMental, 67000 Strasbourg, France

**Keywords:** traumatic reactivation, child and adolescent psychiatry, sleep disorders, somatic symptoms, post-traumatic stress disorders, mass trauma, circadian rhythm disorders

## Abstract

The court trial of the 14th of July 2016 terrorist attack in Nice (France) opened in September 2022 and ended in December 2022. Engaging in court proceedings, whether as a victim or a witness, can lead to a significant risk of traumatic reactivation (i.e., the re-emergence of post-traumatic stress symptoms). The present protocol aimed to improve knowledge of the pathophysiology of traumatic reactivation due to the media coverage of the trial by assessing sleep disturbances and somatic symptoms that could reappear if there is a traumatic reactivation. Method and Analysis: This is a monocentric longitudinal study, with recruitment solely planned at the Nice Pediatric Psychotrauma Center (NPPC). We intended to include 100 adolescents aged 12 to 17 years who were directly or indirectly exposed to the attack and included in the “14-7” program). Assessments began one month before the trial, in August 2022, and were scheduled once a month until the end of the trial. A smartwatch recorded sleep activity. Somatic and PTSD symptoms and sleep were assessed through validated questionnaires. The main analyses comprised the variance and regression analyses of predictors of clinical evolution over time. Ethics and Dissemination: The National Ethics Committee “NORD OUEST III” approved the “14-7” program protocol (number 2017-A02212-51). The specific amendment for this research was approved in April 2022 by the same national ethical committee. Inclusions started in August 2022.

## 1. Introduction

The court trial of the 14th of July 2016 terror attack in Nice was scheduled in 2022 between the 5th of September 2022 and the 15th of January 2023 in Paris and Nice (France) simultaneously. The First Instance Court examined the responsibilities of eight people in addition to the terrorist who died in the attack. At least 865 people or associations were filed as civil parties. The opening and holding of the trial represented possible sources of traumatic reactivation, as well as victim testimonies (from adolescents and caregivers), expert testimonies, the broadcasting of video surveillance of the attack, etc. [1,2].

### 1.1. Traumatic Reactivation

Numerous studies have highlighted that previously extinguished fear responses can reappear in different circumstances. Siegmund and Wotjak (2006) suggest the following hypotheses to explain this phenomenon: a deficit in extinction and habituation, reactivation, or relapse [3]. The extinction and habituation deficit refers to the presence of residual symptoms of post-traumatic stress disorder (PTSD) and a partial extinction of fear responses. Reactivation is the reappearance of symptoms upon confrontation with an event-related stimulus [4,5]. Finally, relapse can be defined as the recurrence of symptoms after remission without reliving new trauma [6].

More recently, other authors postulate that the return of fear responses could result from the coexistence of two memories: the memory of fear and the memory of fear extinction. In this model, the extinction memory inhibits fear memory. The reappearance of PTSD symptoms is then linked to a memory inhibition defect, i.e., the extinction memory is no longer able to inhibit the expression of the fear memory [7,8] (Craske et al., 2014; 2022).

There are few studies concerning the phenomena of relapse and reactivation. Most of the research on this subject is related to pharmacotherapy and provides a frequency of relapse or reactivation among adults with PTSD. The latter is relatively high, with between 20 and 40% of individuals experiencing this within six months after therapy [9], and it rises to 53% five years after therapy [10]. In a 1994 study, Long, Chamberlain, and Vincent found that media images of the Gulf War led to the re-emergence of PTSD symptoms in Vietnam War veterans [11]. Other studies have also shown that experiencing another traumatic event (a surgical operation, accident, or assault) increased traumatic reactivation in refugees [12]). Moreover, intrusion symptoms seem to be predominant in cases of traumatic reactivation, while avoidance is rarer [13].

### 1.2. The Terrorist Attack of July 14th 2016 and Its Consequences for Adolescents

On the 14th of July 2016, in Nice, a terrorist drove a truck into the crowd on the Promenade des Anglais, killing 86 people, including ten children, and injuring around 500 others. In total, 30,000 people attended the National Day fireworks display, including babies, children, adolescents, and their families [14].

Exposure to this attack could have been direct (being injured or seeing others being injured or killed by the truck) or indirect (hearing the attack, being involved in the crowd movement, hearing the police sirens, etc.). Most people on the scene remained confined for several hours in restaurants or stores, where they received alarming and false news on social media of multiple attacks. Additionally, many families were separated after being swept along by the crowd, and some children were separated from their parents [15,16]. These types of exposure can all be potentially traumatic, as described by criteria A of PTSD in the DSM-5 TR.

Many children who consulted the Nice Pediatric Psychotrauma Center (NPPC) after the 14th of July 2016 developed PTSD [17]. Since 2017, the NPPC has been taking care of children and adolescents with various mental disorders, mainly PTSD, following this attack. The treatments offered were specific to the child’s symptoms, ranging from trauma-focused cognitive behavioral therapies (TF-CBT) to eye movement desensitization and reprocessing (EMDR) or therapies such as mindfulness [18]. At the same time, a research program (“14-7” program) was offered to families affected by the attack [19]. This is a longitudinal epidemiological study that focuses on children and adolescents from 0–18 years old who were directly or indirectly exposed to the attack. This study includes psychological assessments of both children and parents (traumatic, depression and anxiety symptoms, quality of life, sociodemographics, IQ) until the child participants reached the age of 25). The preliminary results show that 62% of children and adolescents had PTSD, which did not depend on their age [20].

PTSD is characterized by the following four symptoms: daytime and nighttime reliving (nightmares/night terrors) of the event, cognitive and/or behavioral avoidance, alterations in cognition and mood, and hyperarousal [21] with the presence of sleep disorders [22]. People suffering from PTSD may experience insomnia, difficulty falling asleep, and nightmares with multiple awakenings, all of which impact the quality of life [23]. The latter is common in this condition where up to 90% of children with PTSD have nightmares and severe difficulty falling asleep [23].

### 1.3. Normal Sleep Process

Sleep plays a vital role at biological, cognitive, and emotional levels [24]. Sleep is a state of diminished consciousness that alternates with wakefulness.

Sleep duration in humans varies with age as follows: newborns need a long period of sleep (between 12 and 15 h per day); adolescents begin to reduce their need for sleep, with an average of 9 to 10 h per night; then adults have a reduced sleep duration (6 to 8 h) [25]. Sleep duration is correlated with the development of specific brain areas, notably the hippocampus, in children and adolescents [26].

Human studies show that adequate sleep in children is essential for normal growth and development for the well-being of the mother and family and childhood sleep is associated with significant predictors of adult health [27].

Adolescence is a developmental period that includes immense morphological changes, particularly hormonal, but also major social changes. These changes have an impact on adolescent’s sleep [28]. Adolescents sleep for between 8 and 9 h per night [29,30,31]. They tend to sleep longer on weekends, with an average of 10–11 h of sleep per weekend night [32]. With age, there is an increasing societal demand for early waking and morning activity (school, work), leading to the need to wake up late on weekends [33,34,35,36,37]. Puberty is specifically associated with reductions in the duration of slow-wave sleep [38,39,40] and alterations in circadian rhythm toward a “delayed phase preference” during adolescence [41]. There are also changes in the homeostatic sleep-regulation system that allow a greater tolerance to sleep pressure and, therefore, a delay in falling asleep [41]. These various factors lead to chronic sleep insufficiency [28,42].

Based on a recent concept analysis, sleep quality has been defined as an individual’s self-satisfaction with the different attributes of sleep that comprise sleep efficiency, sleep latency, sleep duration, and waking after sleep onset (WASO) [43]. Sleep efficiency is related to the amount of time sleep is achieved and the total time in bed. Sleep latency is the time needed to fall asleep. Sleep duration is the total sleep time. Finally, WASO is the amount of wake time after the onset of sleep.

### 1.4. Sleep Disorders

Adolescents experience relatively similar sleep disorders as adult populations [44]. Sleep disorders in adolescents are associated with cognitive complaints (memory, concentration, and reasoning) and mood disorders (anxiety, depressed mood, irritability) [45]. Some studies have highlighted the negative impact of reduced sleep time on cognitive functioning. The majority of these studies have been carried out in an adult population and using sleep deprivation. They showed that a short manipulation of sleep led to a decrease in attention [46], executive functions [47], emotional regulation [48], and memory [49,50,51,52]. However, these studies are carried out in a non-ecological way and, therefore, cannot reflect the impact of long-term sleep deprivation, as may be the case in people with chronic sleep disorders. In addition, other authors have shown that children suffering from early sleep disorders are characterized by risky behaviors in adolescence [53].

Although sleep disorders are common during puberty and adolescence, they are often underdiagnosed [54]. Sleep disorders are often associated with anxiety disorders in adolescents [55]. Thus, 25.6% of adolescents with an anxiety disorder also complain of insomnia [56]. This rate seems to differ according to the type of anxiety disorder, ranging from 24.3% in adolescents with a phobia to 42.9% in adolescents with obsessive-compulsive disorder [57]. Another study showed in 106 hospitalized psychiatric patients aged 7 to 16 that 95% of them had moderate to severe sleep problems [58]. Sleep disorders can be found in several pathologies in adolescents, including depression [59], attention deficit hyperactivity disorder (ADHD) [60], impulse control disorders [61], anxiety [62], and bipolar disorder [63], indicating the key role of sleep in adolescent mental health.

Some authors have suggested that changes in adolescent sleeping patterns are tolerable but may be unbearable if aggravated by stressors [64] and increase suicidal ideations [65,66]. Sleep could also influence emotional regulation processes [48]. In adolescents, a reduction in sleep duration can negatively affect mood and disrupt emotional regulation [67,68].

Sleep disorders in adolescents may appear before developing a pathology such as PTSD [69]. After traumatic exposure, sleep disorders are the earliest, most sensitive, and persistent symptoms [69]. They can increase PTSD symptomatology and comorbidities such as depression, suicidal behaviors, general distress, quality of life, and alcohol and drug use [69]. They are also associated with decreased response rates to TF-CBT (CBT) in adolescents [70]. Sleep difficulties are symptoms of PTSD and are multifaceted [71]. In the context of relieving symptoms, the presence of disabling nightmares can lead to multiple awakenings, altering the succession of sleep cycles (light, REM, and deep), thus impacting the adolescent’s sleep quality [72]. In the context of hyperarousal, difficulties in falling asleep are also cited, leading to a reduction in sleep time. In addition, some adolescents may, because of nightmares, be apprehensive about bedtime, which usually leads to an avoidance of sleep [73].

In a study of survivors of the Utoya terrorist attack in 2011, the rate of sleep disorders was 52.4% among survivors. The most common sleep disorders were insomnia (53.3%), frequent nightmares (37.5%), excessive daytime sleepiness (34.4%) and symptoms of obstructive sleep apnea (18.8%). In addition, actigraphy data revealed delayed bedtime, sleep onset, and rising in survivors [74].

### 1.5. Somatic Symptoms

It is known that trouble sleeping is not the only physical difficulty that can affect the daily lives of people with PTSD [75]. Adolescents with traumatic experiences may display somatic symptoms; these symptoms can be chronic with very high intensity and can lead to medical examinations, which are sometimes invasive [76,77]. Adolescents with PTSD have an increased risk of cardiovascular problems, cancer, osteoarthritis, and metabolic disorders in adulthood [78,79]. Several hypotheses have been proposed to explain the association between somatic symptoms and PTSD. For some authors, hyperarousal can mediate the relationship between trauma exposure and somatic symptoms [80]. For other authors, somatic symptoms can be forms of dissociation [81]. Some work highlights an association between the presence of traumatic events in childhood and increased inflammation [82,83]. The inflammatory hypothesis in psychiatric disorders, including mood disorders and schizophrenia, has been increasingly discussed recently in the course of the SARS-CoV-2 pandemic [84]. Furthermore, the combination of trauma and poor family functioning is linked to many forms of somatic symptoms, particularly fatigue, migraines, increased heart rate, nausea, back pain, or muscle pain [85].

### 1.6. Objectives and Hypothesis

This research project offers a valuable opportunity to better understand PTSD reactivation and its impact on sleep in adolescents who witnessed a mass terrorist attack. Hence, the primary objective was to assess, during the court trial, the sleep quality and quantity among a group of adolescents who were directly affected by the attack. The secondary objective was to assess somatic and post-traumatic stress symptoms (PTSS) and monitor their evolution over time.

Since sleep disturbances and somatic symptoms may precede and/or follow (in a variable and unknown time frame) PTSS and acute stress disorder, we hypothesize that sleep disturbances and somatic symptoms may also precede and/or follow traumatic reactivation.

If adolescents present PTSS before the court trial begins, it cannot be considered a traumatic reactivation.

Sleep quality can reflect PTSD symptomatology in adolescents. We use the term PTSS instead of PTSD because the diagnosis is not established by a physician but with a clinical scale. Those with pre-trial sleep difficulties were expected to show an increase in these difficulties over time. Those who experienced traumatic reactivation during the trial were expected to have associated sleep disorders. The main sleep disorders are insomnia, non-restorative sleep, and respiratory problems. The use of connected watches made it possible to obtain data reflecting adolescents’ sleep disturbances. Thus, patients with sleep disturbances, according to the questionnaire, have shorter sleep cycles, lower sleep duration, longer sleep latency, and an increased number of wakes, according to the data collected from connected watches. These difficulties correlate with PTSS.

Somatic symptoms can also be associated with PTSS. Adolescents who presented somatic symptoms before the start of the trial were expected to show an increase in these difficulties over time. Those who experienced traumatic reactivation during the trial were expected to present associated somatic symptoms.

Finally, with regard to the duration of the trial, we expected that different adolescent profiles would emerge. According to the score of the PTSD scale, three profiles are defined: the resilient, who show no traumatic reactivation throughout the legal proceedings; the traumatic reactivators, who show a return of PTSS during the proceedings; and the PTSS, who already show PTSS before the trial begins and maintain their symptoms during the proceedings.

## 2. Methods Section

### 2.1. Participants

A maximum of one hundred adolescents aged 12 to 17 years directly from the “14-7” program were contacted.

The primary endpoint of this study was Bruni et al.’s (1996) Sleep Disorders Screening Scale designed for children older than four years old [86].

The number of necessary subjects was estimated under the assumption that the mean of the total sleep score would be 37.0 and 56.0 for groups PTSS− and PTSS+, respectively, with a common standard deviation of 12.0 [87]. With a type 1 risk of 1.0% and a type 2 risk of 10.0%, a bilateral test, and a dropout rate estimated at 20.0%, we estimate that 15 and 15 patients would be necessary for groups PTSS- and PTSS+, respectively. Based on the “14-7” program, 100 participants were included in this study [20].

#### 2.1.1. Inclusion Criteria

Adolescents who were children in 2016 at the time of the terrorist attack (between 7 and 12 years of age) and who were included in the “14-7” program were included in this study. They had a good understanding of the French language (native French-speaking or fluent in French); adolescents agreed to participate (assent collection) whose parents agreed to participate in the study (informed consent collection).

#### 2.1.2. Non-Inclusion Criteria

Adolescents with a moderate intellectual disability (IQ less than 50);

Adolescents deprived of liberty by judicial or administrative decisions;

Participation in another work of research with an exclusion period.

#### 2.1.3. Exclusion and Study Exit Criteria

A temporary or definitive interruption of the study, depending on the severity and circumstances involved, can be made in the case of the following:

A simple request from the adolescent or their parents (the interruption of participation or withdrawal of consent);

Difficulties with the equipment provided (a synchronization problem, lost watch, broken watch, etc.); An inability to comply with the study procedures and instructions presented and explained at the time of inclusion.

### 2.2. Ethical Consideration, Funding, and Registration

This study on the risks of traumatic reactivation during the 14th of July 2016 Nice terrorist attack court trial was filed as an amendment to the “14-7” program. The amendment was approved on 2 April 2022 by the National Ethics Committee “NORD OUEST III” (number 2017-A02212-51, MSn°06). This research received no specific grant from any funding agency in the public, commercial, or not-for-profit sectors. The “14-7” program was also registered with ClinicalTrials.gov (number NCT03356028).

After being informed about the study, patients, and their caregivers were asked to sign the assent and the informed consent. Actual enrolment started on the 5th of August 2022.

### 2.3. Study Design and Procedures

After receiving explanations about the study’s objectives, details about the study-specific procedures, and signing the informed consent, a smartwatch was provided to each participant to record their sleep. Sleep was recorded continuously every first week of each month between August and January. The Fitbit application was downloaded onto the child’s cell phone with the consent of the participant and his or her parents.

The evaluator showed the teenager how to charge the watch. The watch was then placed on the teenager’s wrist. In the event of an adverse event (an allergy, for example), precautionary measures were explained to the teenager and his or her parents (remove the watch, rinse off, apply a soothing ointment, consult a general practitioner).

In parallel, online questionnaires on PTSD, somatic symptoms, and sleep were completed (see Figure 1). In the first session, participants completed the questionnaires face-to-face with the evaluator. Subsequent sessions were conducted remotely via the Internet. The teenager received an email reminding him or her to fill in the various scales. If they so wished, a visual consultation could be carried out to help them fill in the questionnaires.

If the court trial dates change, the study will adjust to the legal procedure by postponing and/or increasing the duration of sleep and clinical assessments.

### 2.4. Study Measures

#### 2.4.1. Primary Endpoint

The primary endpoint of this study is the French version of the Bruni Sleep Disorders Screening Scale designed for children older than four years old [86,87,88,89]. This scale is a validated self-report questionnaire used in research and clinical practice to assess and screen for various sleep disorders. This scale consists of a series of questions covering different aspects of sleep, including a global sleep quality score and sub-indices assessing insomnia, parasomnias, respiratory problems, non-restorative sleep, and excessive daytime sleepiness. In this questionnaire, the child has to answer questions like “Do you go to bed reluctantly?” “Do you have trouble falling asleep?” using a 5-point Lickert scale ranging from Never = 1 to Always = 5. This self-questionnaire was fulfilled during the month before the start of the court trial and then monthly until the end of the court trial. The cut-off scores calculated in the validation study (Pathological threshold: total score > 70) of the scale were used to define the following two groups: Sleep Disturbances vs. No Sleep Disturbances.

#### 2.4.2. Secondary Endpoints

Smartwatches have shown reliability in assessing nocturnal activity in adults (sleep phases, presence of nightmares, and wakefulness phases) [90,91]. These wearable devices are equipped with sensors providing reliable and objective measurements of sleep-related parameters.

Sleep quality is assessed by the onset and offset of sleep, the sleep duration per night, the number of awakenings per night, as well as the duration of each sleep stage (REM, light, and deep).

PHQ-13: This questionnaire is based on the PHQ-15, Patient Health Questionnaire-15 (without the two adult items on menstruation and sexuality), which assesses the frequency of somatic symptoms over a week [92]. The PHQ-15 is a self-reported questionnaire used to assess somatic symptoms in clinical practice, specifically focusing on somatic symptoms potentially associated with emotional or psychological distress. In adults, PHQ-15 is sufficiently validated (primary care), and a link between PHQ-15 and PTSD is also demonstrated [93]. The questionnaire consists of 15 items assessing the presence and severity of the following somatic symptoms: stomach pain, back pain, joint pain, migraines or headaches, chest pain, dizziness, fainting spells, accelerated heart rate, shortness of breath, constipation or diarrhea, nausea, gas or indigestion, fatigue, and sleep disorders. The participant answers the questions using a 3-point Likert scale (Not bothered at all = 0 to Very bothered = 2).

Child PTSD Checklist (CPC) child version (CPC-C): the CPC is updated for DSM-5 and is available in two versions, a parent/caregiver version (CPC-P) and a child-report version (CPC-C). Only the CPC-C, i.e., the child self-report, was used for this study. This self-questionnaire is divided into three parts. The first part includes a list of traumatic events. Then, the participant must answer 20 questions to evaluate the intensity of their PTSD symptoms over the last 15 days. The threshold score is greater than or equal to 20. The questions assess the four main symptoms of PTSD. For example, “Have you had more nightmares since the event?” “Have you had trouble concentrating since the event?”. Finally, the last part includes the evaluation of functional impairment, with six questions concerning family functioning or social functioning. Our team validated the CPC child version in 2021 in a French pediatric population to assess Post-Traumatic Stress Symptoms (PTSS) in youths based on the DSM 5 criteria [94]. Cut-off scores were used to define two groups: PTSS+ vs. PTSS−. We chose PTSS instead of PTSD because the diagnosis was not established by a physician, only through using the scale.

### 2.5. Data Collection Process

All anonymized data were collected online using REDCap (Research Electronic Data Capture), a secured web-based application designed for data collection, management, and the analysis of research studies and online surveys [95,96]. At the end of the last runs, the database was downloaded and securely stored at the University Department of Child and Adolescent Psychiatry (Lenval Hospital—Nice, FRANCE). Sleep data were collected using FitBit Inspire 2—connected watches. Data extraction, transformation, and load (ETL process) were all conducted using a new tool designated for the research called ADETfs (unpublished data, under submission).

The collected watch sleep data consists of the lengths and number of sleep phases (light deep, REM), sleep duration, sleep efficiency, sleep latency, and WASO. Activity data were also collected, including the number of minutes at different activity levels.

### 2.6. Statistical Analysis

To test the hypotheses on sleep disorders assessed by the self-questionnaire, descriptive analyses were carried out at each time point on the overall score, as well as on the sub-scores on the whole sample, but also according to the presence or absence of PTSS (PTSS+ group vs. PTSS− group). Pearson’s correlations were performed between the PTSS symptoms score and sleep disorders. Repeated-measures ANOVA was performed between the overall questionnaire scores, as well as sub-scores according to trauma symptoms (PTSS+ vs. PTSS−) and the assessment time.

To test these hypotheses on somatic symptoms, descriptive analyses were carried out at each time point on the overall score (intensity of somatic symptoms) and on the number of somatic symptoms for all the samples and also according to the presence or absence of PTSS (PTSS+ group vs. PTSS− group). Pearson’s correlations were performed between the somatic symptom intensity score and number of somatic symptoms score with the sleep scale scores and PTSS symptoms scores. Repeated-measures ANOVAs were performed between the intensity, the number of somatic symptoms as a function of trauma symptoms (PTSS+ vs. PTSS−), and assessment time.

To test these exploratory hypotheses on the use of connected watches, correlations were carried out between data extracted from smartwatches (onset and off-set times of sleep, total sleep duration, number of awakenings, sleep stages durations) and the sleep self-questionnaire. ANOVAs were performed on the sleep data collected from watches according to the overall score obtained on the sleep questionnaire (sleep disturbances group vs. no sleep disturbances group). To test the hypothesis of the influence of PTSS, repeated-measure ANOVAs were performed on the data collected from the watches as a function of PTSS (PTSS+ vs. PTSS− group) and the evaluation time factor.

Finally, multiple regressions were then performed to test if sleep measures and somatic symptoms could predict traumatic reactivation symptoms over time.

## 3. Outcomes

Each clinician who provided care and support to victims, as well as all researchers in the field, knows that the court trial time is a crucial phase in the victims’ lives for numerous reasons, among them seeking justice or closure and healing. While extensive research has been conducted on trauma, there is a surprising dearth of scientific data on the risk of traumatic reactivations in pediatric populations. Considering that children and adolescents may respond differently than adults to traumatic experiences, including the reappearance of symptoms upon confrontation with event-related stimuli, the lack of studies and scientific data jeopardizes our ability to develop optimal therapeutic strategies and preventive interventions.

Recently, the 13th of November 2015 terrorist attacks in Paris court trial revealed numerous and diverse expectations and emotional implications for the victims and their families.

Sleep and somatic symptoms can be easily assessed by all clinicians, physicians, or psychologists, by the children’s parents, and, of course, by the adolescents themselves. Therefore, these assessments could be used as non-invasive and ecological markers of the risks of traumatic reactivations in the cases of legal proceedings. Legal proceedings and court trials are known to provoke anxiety because they constitute critical reminders of the traumas experienced by the victims. Although the court trial of the 14th of July 2016 terrorist attack in Nice represents a specific situation, the data obtained could be transposed to other types of trauma that generate legal proceedings.

In addition, psychological and psychiatric assessments need to be adapted and modernized, especially for young people. The use of technological tools accessible to a great number of individuals, such as connected watches, enables adolescents to become active players in their own care. Further studies are needed to include this type of use of new technologies in routine care.

In a similar way, it is important in the future to take into account the care pathways followed by children and adolescents to better understand the mechanisms of traumatic reactivation and to be able to prevent them.

## 4. Conclusions

Children and adolescents in Nice were doubly affected by the attack on the 14th of July 2016. Indeed, many witnessed the attack directly [14,15,97,98,99] and were also indirectly exposed, facing caregivers and later manifesting anxiety disorders or PTSD [19,20].

The “14-7” program, a longitudinal epidemiological study evaluating the impact of a terrorist attack on the development of children and adolescents, has followed 689 participants over the past five years. Among them, 182 were aged 7 to 12 years at the time of the attack and could, therefore, participate in the study on sleep disorders and somatic symptoms in the context of this trial.

This study could provide a non-invasive, reproducible, and easy-to-implement way of improving our knowledge of the pathophysiology of traumatic reactivation during a trial by assessing sleep quality and quantity, somatic and PTSD symptoms in a cohort of adolescents previously affected by a terrorist attack. In addition, this study should pave the way for preventive strategies and specific care to manage or limit traumatic reactivation during legal proceedings, such as psychoeducation on legal proceedings (with age-related tools: comics, ludic applications, etc.) and the risks of traumatic reactivation, peer support groups, and individual consultations.

## Figures and Tables

**Figure 1 healthcare-11-02953-f001:**
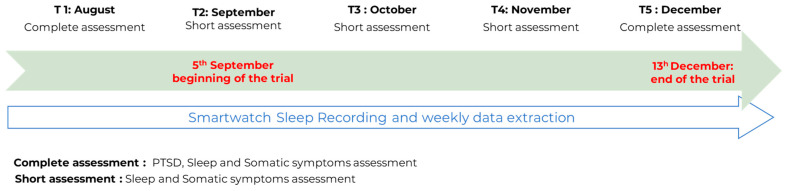
Sleep and clinical assessments throughout the court trial.

## Data Availability

The data presented in this study are available on request from the corresponding author.

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
