# Peer review of "July 14th 2016 Nice Terrorist Attack Court Trial: A Protocol on Sleep Quality and Somatic Symptoms as Markers of Risk for Traumatic Reactivation in Adolescents Exposed to This Attack"

_healthcare, 2023, doi:10.3390/healthcare11222953_

Round 1

Reviewer 1 Report (Previous Reviewer 1)

Comments and Suggestions for Authors

There are significant improvements to this manuscript and the overall quality has improved. Authors appeared to have worked hard to improve the manuscript.

However there are a few minor errors to be addressed still:

Line 179 - "Iis" appears to be a typo

Line 180 - there is an extra bracket inserted

Line 203 - spacing error in text

Line 212 - authors mention that sleep quality should reflect PTSD symptomatology. One of the DSM criteria are night terrors. Are authors alluding that the DSM falls short in highlighting this symptom? Same for Line 221 - is it that authors state DSM falls short or they expect to see an increase in symptoms or both? 

Line 239-257 - author appear to have written this paragraph in a bullet point form. This is best presented in paragraph from.

Line 267, 304, 347, 354 - formatting error

Line 356 - "anovas" is an acronym 

Comments on the Quality of English Language

Please see prior comments 

Author Response

Comments and Suggestions for Authors

There are significant improvements to this manuscript and the overall quality has improved. Authors appeared to have worked hard to improve the manuscript.

However there are a few minor errors to be addressed still:

Line 179 - "Iis" appears to be a typo

Line 180 - there is an extra bracket inserted

Line 203 - spacing error in text

Line 212 - authors mention that sleep quality should reflect PTSD symptomatology. One of the DSM criteria are night terrors. Are authors alluding that the DSM falls short in highlighting this symptom? Same for Line 221 - is it that authors state DSM falls short or they expect to see an increase in symptoms or both?

Line 239-257 - author appear to have written this paragraph in a bullet point form. This is best presented in paragraph from.

Line 267, 304, 347, 354 - formatting error

Line 356 - "anovas" is an acronym

We would like to thank the reviewer for their comments. We have corrected all typos and formatting errors mentioned. Regarding the DSM, the DSM-5-TR PTSD criterias includes trouble falling asleep , night terrors, and nightmares. We intend to investigate if these disturbances appear during the trial which would, in addition to the PTSS scale (CPC), indicate there is trauma reactivation.

Reviewer 2 Report (Previous Reviewer 2)

Comments and Suggestions for Authors

The authors made relevant changes in order to address comments left by me on the previous version.

My concerns have been adequately addressed. 

I am still concerned about what new knowledge will be uncovered by this research. The research question for this article has already been answered by prior research. 

see this - 

1. Thierrée S, Richa S, Brunet A, et al. Trauma reactivation under propranolol among traumatized Syrian refugee children: preliminary evidence regarding efficacy. Eur J Psychotraumatol. 2020;11(1):1733248. Published 2020 Mar 3. doi:10.1080/20008198.2020.1733248

2. Kendil N. « Quand il faut donner du sens au non-sens du trauma » L’intervention psychologique auprès des victimes de la fusillade du 29 janvier 2017 à Québec – dimension multiculturelle ["When we have to make sense to the nonsense of trauma" Psychological intervention among shooting victims of January 29, 2017. Multicultural perspective]. Sante Ment Que. 2019;44(1):11-30.

Comments on the Quality of English Language

Apart from a few typing errors, the language is easy to read and understand. 

Author Response

Comments and Suggestions for Authors

The authors made relevant changes in order to address comments left by me on the previous version.

My concerns have been adequately addressed.

I am still concerned about what new knowledge will be uncovered by this research. The research question for this article has already been answered by prior research.

see this - 1. Thierrée S, Richa S, Brunet A, et al. Trauma reactivation under propranolol among traumatized Syrian refugee children: preliminary evidence regarding efficacy. Eur J Psychotraumatol. 2020;11(1):1733248. Published 2020 Mar 3. doi:10.1080/20008198.2020.1733248

  1. Kendil N. « Quand il faut donner du sens au non-sens du trauma » L’intervention psychologique auprès des victimes de la fusillade du 29 janvier 2017 à Québec – dimension multiculturelle ["When we have to make sense to the nonsense of trauma" Psychological intervention among shooting victims of January 29, 2017. Multicultural perspective]. Sante Ment Que. 2019;44(1):11-30.

We would like to thank the reviewer for their comments. In the first study that the reviewer mentioned, authors investigate the efficacity of a pharmacological treatment on trauma symptoms for refugees. Trauma reactivation is induced voluntarily for the purpose of their study. The second one investigates the efficiency of a psychological treatment right after a shooting attack. Despite the interest of these articles, our research questions do not include any treatment hypothesis.
Our research intends to investigate how sleep is affected during trauma reactivation due to a court trial 7 years after a mass terrorist attack. The protocol has two main interests, the primary hypothesis that trauma reactivation has indeed an effect on sleep quality. The second one is the use of new digital markers extracted from smartwatches and if these fine and objective measures are reliable compared to clinical scales.

Reviewer 3 Report (New Reviewer)

Comments and Suggestions for Authors

The study was read carefully.

 The introduction was sufficient.

 Q1: Method, line 236

How to calculate the sample size?

Q2: line 269

How to make sure the smartwatch is applied adequately?

Q3: line 297

The cut-off scores will be to define two groups.

How the scores to define these two groups?

Q4: Statistical analysis

Repeated ANOVAS

Why not regression?

Q5. Line 368,

Latent profile analysis (LPA) and confirmatory factor analysis (CFA)

Latent profile analysis: continuous variable vs. categorical variable

Why not use the regression model?

Q6. Why not no Result section?

This is the major concern.

The author might present the results from this study, such as basic data, ANOVA results, results for LPA, CFA and even regression model.

All over all, this article is more like a project than a research.

Author Response

Comments and Suggestions for Authors

The study was read carefully.

The introduction was sufficient.

 Q1: Method, line 236

How to calculate the sample size?

Based on previous studies using the Bruni et al. (1996) Sleep Disorders Screening Scale designed for children older than four years old, who defined mean scores of 25 for control populations, with varying standard deviations (mean 25) [86-87], the minimum sample size required to test our hypothesis was estimated at 68 participants. To prevent experimental attrition, we intended to include 100 participants. 

We have added the last sentence into the participants  paragraph section.

Q2: line 269

How to make sure the smartwatch is applied adequately?

We have worked with a fellow engineer who created a program to extract fitibit data automatically. His program can include errors linked to synchronization data missing. With this, we could identify participants who did not wear the smartwatches or could not synchronize the data with their smartphones.

Q3: line 297

The cut-off scores will be to define two groups.

How the scores to define these two groups?

The cut-off scores were extracted from the validation study. We have added this information Line 297. 

The cut-off scores calculated in the validation study (Pathological threshold: total score >70) of the scale, will be used to define two groups: Sleep Disturbances vs No Sleep Disturbances.

Q4: Statistical analysis

Repeated ANOVAS

Why not regression?

In our protocol, we will have both quantitative and qualitative data, therefore we prefer ANOVAs to test our hypothesis in this context.

Q5. Line 368,

Latent profile analysis (LPA) and confirmatory factor analysis (CFA)

Latent profile analysis: continuous variable vs. categorical variable

Why not use the regression model?

The reviewer is right. We have changed the paragraph to: “ Finally, multiple regressions will then be performed to test if sleep measures and somatic symptoms can predict traumatic reactivation symptoms over time.”

Q6. Why not no Result section?

This is the major concern.

The author might present the results from this study, such as basic data, ANOVA results, results for LPA, CFA and even regression model.

 All over all, this article is more like a project than a research.

The study is in fact a study protocol. The data has therefore not yet been analysed. We are waiting for the protocol to be published before we can process the data.

Reviewer 4 Report (New Reviewer)

Comments and Suggestions for Authors

This manuscript represents the protocol for the research on relapse of PTSD in the terrorism trial in adolescents. Such study in adolescents is rare. It would be important to clarify the relationship between sleep and somatic symptoms and PTSD recurrence. However, there are a number of serious problems with this paper, including insufficient comparison of past and planned studies and a lack of clearly defined research methodology. I found it a bit difficult to know how to evaluate this work. Authors should also clarify and correct the following points

Introduction

The purpose of this study was to be thought the relationship between PTSD recurrence and sleep disturbances and physical symptoms. However, the Introduction is about sleep, sleep disturbances, and somatic symptoms. These are redundant because they are general knowledge. What we know so far should be stated in order, such as the occurrence of PTSD due to terrorism and other incidents in adolescents, the relationship between PTSD and sleep and its disturbances, PTSD and somatic symptoms, and PTSD relapse (especially reported in adolescents). The authors should then state their hypothesis. The introduction should conclude with the objectives of the study.

Methods

Participants

The description of the sample size is inadequate. How can 100 examples clarify the purpose of this study? You should show statistical calculations or citations from previous work.

Inclusion criteria, non-inclusion criteria, exclusion and study exit criteria must be included in participants or use 2.1.2, 2.1.3, 2.1.4

According the statement in Introduction “Finally, with regard to the duration of the trial, different adolescent profiles should emerge. The resilient, who show no traumatic reactivation throughout the legal proceedings; the traumatic reactivators, who will show a return of PTSS during the proceedings; and the PTSS, who already show PTSS before the trial begins.”, authors should define these participants.

Study design and procedures

When does this study start and end? How long is the observation period? Please provide the timeframe for enrollment. The trial run until January 2023, and it ended when this paper was submitted. Does this mean that this is a retrospective study? If so, the protocol should be in the past tense.

The outcomes are the primary and second endpoints.

Study Measures must state all measures of the study and the details.

Conclusion

Discuss the merits of this study in comparison to previous studies.

What does this study reveal? How will it contribute to society?

Limitations

Not normally required in protocol papers.

L114, 203, 304, 361: There is extra space.

Programme 14-7, 14/7 program, 14-7 Research Program, "14-7" research Program, "14-7" Program Do they all mean the same thing? Since I am not familiar with this study, please provide a brief description (subjects, duration of study, measurements, protocol, etc).

L297: Please define the cut-off point

The word “PTSS” means “PTSD”?

L179: The word “Iis” means “Its”?

L356, 365: The word “Anovas” means “ANOVAs”?

L64: Posttraumatic stress disorder found in L58.

L89, 153, 332: Usually described in order of spelling (abbreviation).

L 88 & 165: trauma-focused cognitive behavioral therapies (TF-CBT) and PTSD-focused cognitive behavioral therapy (CBT) are same?

L267: Please break line.

Comments on the Quality of English Language

There are grammatical errors and formatting problems.

Author Response

Comments and Suggestions for Authors

This manuscript represents the protocol for the research on relapse of PTSD in the terrorism trial in adolescents. Such study in adolescents is rare. It would be important to clarify the relationship between sleep and somatic symptoms and PTSD recurrence. However, there are a number of serious problems with this paper, including insufficient comparison of past and planned studies and a lack of clearly defined research methodology. I found it a bit difficult to know how to evaluate this work. Authors should also clarify and correct the following points

Introduction

The purpose of this study was to be thought the relationship between PTSD recurrence and sleep disturbances and physical symptoms. However, the Introduction is about sleep, sleep disturbances, and somatic symptoms. These are redundant because they are general knowledge. What we know so far should be stated in order, such as the occurrence of PTSD due to terrorism and other incidents in adolescents, the relationship between PTSD and sleep and its disturbances, PTSD and somatic symptoms, and PTSD relapse (especially reported in adolescents). The authors should then state their hypothesis. The introduction should conclude with the objectives of the study.

We thank the reviewer for their comment. We have shortened the introduction, specially the normal sleep section and general knowledge concerning sleep. Regarding the objectives of the study, a specific paragraph (1.6.) is dedicated to it in the introduction.

Methods

Participants

The description of the sample size is inadequate. How can 100 examples clarify the purpose of this study? You should show statistical calculations or citations from previous work.

The reviewer is right. We modified the values to calculate the sample size using the french validation of the leep scale (Putois et al., 2017). We changed the paragraph to:

“The number of necessary subjects was estimated under the assumption that the mean of the total sleep score would be 37.0 and 56.0 for groups PTSS- and PTSS+ respectively with a common standard deviation of 12.0 based on Putois et al., 2018 validation study. With a type 1 risk of 1.0% and a type 2 risk of 10.0%, a bilateral test and a drop out rate estimated at 20.0%, we estimate that 15 and 15 patients would be necessary for groups PTSS- and PTSS+ respectively. Based on the “14-7” program, 100 participants could be included in this study (Askenazy et al. 2023).  “

Inclusion criteria, non-inclusion criteria, exclusion and study exit criteria must be included in participants or use 2.1.2, 2.1.3, 2.1.4

We have followed the reviewer’s suggestion for the participants section.

According the statement in Introduction “Finally, with regard to the duration of the trial, different adolescent profiles should emerge. The resilient, who show no traumatic reactivation throughout the legal proceedings; the traumatic reactivators, who will show a return of PTSS during the proceedings; and the PTSS, who already show PTSS before the trial begins.”, authors should define these participants.

We added a sentence to better explain the three groups : « According to the score of PTSD scale, three profiles are defined: the resilient, who show no traumatic reactivation throughout the legal proceedings; the traumatic reactivators, who will show a return of PTSS during the proceedings; and the PTSS, who already show PTSS before the trial begins and maintains the symptoms during the proceedings.”, authors should define these participants ».

Study design and procedures

When does this study start and end? How long is the observation period? Please provide the timeframe for enrollment. The trial run until January 2023, and it ended when this paper was submitted. Does this mean that this is a retrospective study? If so, the protocol should be in the past tense.

We added a figure to clarify the timeframe. As this is a prospective study and even though the inclusions have been completed, we believe that the past is not necessary.

The outcomes are the primary and second endpoints.

Study Measures must state all measures of the study and the details.

In the study measures section, we have added the cut-offs used for each questionnaire.

Conclusion

Discuss the merits of this study in comparison to previous studies.

What does this study reveal? How will it contribute to society?

Our research intends to investigate how children and adolescents are affected by a court trial 7 years after a mass terrorist attack. The protocol has two main interests, the primary hypothesis that trauma reactivation has indeed an effect on sleep quality and somatic symptoms. The second one is the use of new digital markers extracted from smartwatches and if these fine and objective measures are reliable compared to clinical scales. This is the first step on the evaluation of the impact of legal proceedings on child and adolescents mental health. We hope that future researches emerge in this thematic with quantitative and qualitative approaches to better understand this topic.

Limitations
Not normally required in protocol papers.

L114, 203, 304, 361: There is extra space.

Programme 14-7, 14/7 program, 14-7 Research Program, "14-7" research Program, "14-7" Program Do they all mean the same thing? Since I am not familiar with this study, please provide a brief description (subjects, duration of study, measurements, protocol, etc).
We have harmonized the paper and the program name. We have added more explanation in the introduction section: “The study includes psychological assessments of both children and parents (traumatic, depression and anxiety symptoms, quality of life, socio demographics, IQ) until the children participants reach the age of 25).”

L297: Please define the cut-off point.

The cut-off scores calculated in the validation study (Pathological threshold: total score >70) of the scale, will be used to define two groups: Sleep Disturbances vs No Sleep Disturbances.

The word “PTSS” means “PTSD”?

We chose PTSS (Post Traumatic Stress Symptoms) instead of PTSD because the diagnosis will not be established by a physician using gold standard interviews but only using the specific scale.

L179: The word “Iis” means “Its”?

L356, 365: The word “Anovas” means “ANOVAs”?

L64: Posttraumatic stress disorder found in L58.

L89, 153, 332: Usually described in order of spelling (abbreviation).

L 88 & 165: trauma-focused cognitive behavioral therapies (TF-CBT) and PTSD-focused cognitive behavioral therapy (CBT) are same?

We thank the reviewer, it is done. The two therapies are the same. We modified PTSD focused cognitive behavioral theraby to TF-CBT.

L267: Please break line.

It is done.

Reviewer 5 Report (New Reviewer)

Comments and Suggestions for Authors

This study presents a unique opportunity to study the epidemiology of PTSD in children and adolescents who were exposed to a horrific terrorist attack.  The authors do a good job in their introduction setting the stage for why their DVs should include both somatic symptoms, PTSD, and sleep disturbances over time after exposure to a serious trauma. The authors present the symptoms and possible trajectories of children and adolescents with PTSD and present past literature that shows that sleep and somatic symptoms may wax and wane over time and resurge with the presentation of new event-related stimuli, like the trial that occurred years later. They do a nice job presenting long-term problems with untreated PTSD in the areas of psychiatric, medical, and sleep disturbances.  This is an opportunity to study the trajectory of these symptoms over time in a sample of children and adolescents, especially in the understanding of how something like a trial which can reactivate the trauma exposure. 

The plan for analysis looks good. The correlations and repeated ANOVAs are appropriate. I'm wondering if all of the participants had multiple therapies or if some had only one type of therapy allowing the authors to potentially evaluate the buffering effect of different types of therapies? That might be a nice addition to the analysis plan.

Only one typo that I caught... line 179 His = It is.

While these types of events are horrific, they present an opportunity for us to gain more knowledge and insight into PTSD, which in the end may present more treatment ideas.

Author Response

Comments and Suggestions for Authors

This study presents a unique opportunity to study the epidemiology of PTSD in children and adolescents who were exposed to a horrific terrorist attack.  The authors do a good job in their introduction setting the stage for why their DVs should include both somatic symptoms, PTSD, and sleep disturbances over time after exposure to a serious trauma. The authors present the symptoms and possible trajectories of children and adolescents with PTSD and present past literature that shows that sleep and somatic symptoms may wax and wane over time and resurge with the presentation of new event-related stimuli, like the trial that occurred years later. They do a nice job presenting long-term problems with untreated PTSD in the areas of psychiatric, medical, and sleep disturbances.  This is an opportunity to study the trajectory of these symptoms over time in a sample of children and adolescents, especially in the understanding of how something like a trial which can reactivate the trauma exposure. 

The plan for analysis looks good. The correlations and repeated ANOVAs are appropriate. I'm wondering if all of the participants had multiple therapies or if some had only one type of therapy allowing the authors to potentially evaluate the buffering effect of different types of therapies? That might be a nice addition to the analysis plan.

The participants in this study were followed in the centre by psychologists using EMDR, CBT or body or relationship therapies.

Initially, we had not planned to analyze the impact of the type of therapy on the risk of traumatic reactivation, but this question would be of great interest in future research. We have therefore added a sentence in the outcomes about future research. « In a similar way, it would be important in future to take into account the care pathways followed by children and adolescents, to better understand the mechanisms of traumatic reactivation and to be able to prevent them ».

Only one typo that I caught... line 179 His = It is.

We thank the reviewer, correction is done.

While these types of events are horrific, they present an opportunity for us to gain more knowledge and insight into PTSD, which in the end may present more treatment ideas.

Reviewer 6 Report (New Reviewer)

Comments and Suggestions for Authors

Healthcare-2643873

Thank you for the opportunity to review the manuscript titled: “July 14th, 2016, Nice Terrorist Attack Court Trial: A Protocol on Sleep Quality and Somatic Symptoms as markers for Risk for Traumatic Reactivation in Adolescents Exposed to This Attack.”

This manuscript outlines the protocol of a study in program that will provide rare and noteworthy longitudinal data of trauma (the 2016 Nice terrorist attack)-exposed adolescents experiencing an event (the related trial) that is likely to lead to a return/exacerbation of PTSD symptoms.

While the aims are commendable, there are several other opportunities for improvement that I have noted below. I believe the paper would be strengthened by addressing these issues more fully. 

1.      Lines 62-71 discusses the idea of re-emergence of PTSD symptoms. It might be worth mentioning the inhibitory retrieval model of extinction learning from Michelle Craske, which addresses the issue of trauma associations not going away (through exposure), but rather, stronger associations are formed: https://www.sciencedirect.com/science/article/pii/S0005796722000407

2.      The term “neurovegetative overactivation” was unfamiliar to me and might be unknown to many readers. It appears to refer to what we often call “hyperarousal.” I’d suggest either using the standard term or clarifying how this one is different (and/or why it’s used instead).

3.      I thought section 1.3 on normal sleep processes could be substantially reduced. Lines 102-108 seem unnecessary, for instance. The paragraph at line 114 also seems like it should be moved, possibly to the end of the section.

4.      Line 163: “It” should be “they” (sleep disorders).

5.      Line 171-172: It sounds like this is referring to avoidance of sleep and I think it would be useful to frame it as such directly.

6.      Line 179 seems to have a typo for the first word. Should it be “It is”?

7.      Line 205: It’s worth clarifying that sleep disturbance is a symptom of PTSD itself. The way this sentence is worded seems to imply that the sleep disturbance is a separate thing that could come before.

8.      Line 210: The acronym “PTSS” appears here and comes up afterward throughout the article, but isn’t defined. If this is a different word for PTSD I would recommend picking one to use consistently.

9.      Line 211: I wondered if there would be news coverage or discussion of the trial before it begins – would these not also present risks of traumatic reactivation?

10.  Line 284: From the study design in Figure 1, it appears that there weren’t monthly PTSD symptom questionnaires. Was there any reason for that? There’s a bidirection influence between sleep disturbance and other PTSD symptoms that can sometimes worth together to make treatment challenging (i.e., patient isn’t getting enough sleep to effectively engage in treatment and take in new learning, but PTSD symptoms are too severe to treat the insomnia while ignoring PTSD). It seems like one major benefit of the study could have been to get novel data on the timing of which symptoms re-emerge first, and examine how they lead to subsequent symptoms.

11.  Lines 408-409: I wonder if more specific information could be added here for what strategies might emerge from the results. It seems like there is already a pretty good sense that the trial will lead to a recurrent of PTSD symptoms.

12.  In terms of limitations, the different connection each participant had with the attack seems like it would have been worth examining. For instance, someone who’s life was in danger might have a different reaction that someone who was indirectly impacted, and it would be valuable to see data on this difference.

Author Response

Comments and Suggestions for Authors

Thank you for the opportunity to review the manuscript titled: “July 14th, 2016, Nice Terrorist Attack Court Trial: A Protocol on Sleep Quality and Somatic Symptoms as markers for Risk for Traumatic Reactivation in Adolescents Exposed to This Attack.”

This manuscript outlines the protocol of a study in program that will provide rare and noteworthy longitudinal data of trauma (the 2016 Nice terrorist attack)-exposed adolescents experiencing an event (the related trial) that is likely to lead to a return/exacerbation of PTSD symptoms.

While the aims are commendable, there are several other opportunities for improvement that I have noted below. I believe the paper would be strengthened by addressing these issues more fully.

  1. Lines 62-71 discusses the idea of re-emergence of PTSD symptoms. It might be worth mentioning the inhibitory retrieval model of extinction learning from Michelle Craske, which addresses the issue of trauma associations not going away (through exposure), but rather, stronger associations are formed: https://www.sciencedirect.com/science/article/pii/S0005796722000407

We thank the reviewer for this valuable reference. We have added this paragraph to the introduction in the 1.1. traumatic reactivation section: :  More recently, other authors postulate that the return of fear responses could result from the coexistence of two memories: the memory of fear and the memory of fear extinction. In this model, the extinction memory inhibits the fear memory. The reappearance of PTSD symptoms would then be linked to a memory inhibition defect, i.e. the extinction memory would no longer be able to inhibit the expression of the fear memory (Craske et al., 2014; 2022).

  1. The term “neurovegetative overactivation” was unfamiliar to me and might be unknown to many readers. It appears to refer to what we often call “hyperarousal.” I’d suggest either using the standard term or clarifying how this one is different (and/or why it’s used instead).

The reviewer is right. We have changed the term to hyperarousal.

  1. I thought section 1.3 on normal sleep processes could be substantially reduced. Lines 102-108 seem unnecessary, for instance. The paragraph at line 114 also seems like it should be moved, possibly to the end of the section.
  2. Line 163: “It” should be “they” (sleep disorders).
  3. Line 171-172: It sounds like this is referring to avoidance of sleep and I think it would be useful to frame it as such directly.

We have changed the sentence: In addition, some adolescents may, because of nightmares, be apprehensive about bedtime, which usually leads to an avoidance of sleep [72].

  1. Line 179 seems to have a typo for the first word. Should it be “It is”?
  2. Line 205: It’s worth clarifying that sleep disturbance is a symptom of PTSD itself. The way this sentence is worded seems to imply that the sleep disturbance is a separate thing that could come before.

We modified the sentence to: « Sleep difficulties are symptoms of PTSD and multifaceted [70] ». 

  1. Line 210: The acronym “PTSS” appears here and comes up afterward throughout the article, but isn’t defined. If this is a different word for PTSD I would recommend picking one to use consistently.

We use the term PTSS instead of PTSD because because the diagnosis will not be established by a physician but with a clinical scale. We added this sentence to both objectives and hypothesis section and Method section.

  1. Line 211: I wondered if there would be news coverage or discussion of the trial before it begins – would these not also present risks of traumatic reactivation?

The reviewer is right. Risks of traumatic reactivation are high due to the media  attention for this large-scale event. Anniversary commemorations are known to reactivate symptoms for some of our patients as well as other news events (terrorist attacks, wars, etc.). We decided to include patients one month before the trial in our methodology, which is one month after the 14th July.  It is a limitation of our study that we will take into account when data will be processed.

  1. Line 284: From the study design in Figure 1, it appears that there weren’t monthly PTSD symptom questionnaires. Was there any reason for that? There’s a bidirection influence between sleep disturbance and other PTSD symptoms that can sometimes worth together to make treatment challenging (i.e., patient isn’t getting enough sleep to effectively engage in treatment and take in new learning, but PTSD symptoms are too severe to treat the insomnia while ignoring PTSD). It seems like one major benefit of the study could have been to get novel data on the timing of which symptoms re-emerge first, and examine how they lead to subsequent symptoms.

The reviewer is right. The reason why we did not systematically include the PTSS scale, is due to how patients can react to the scale. The monthly appointments scheduled after the first visit are supposed to be at distance using redcap self-questionnaires format. The PTSD scale needs to be filled with the presence of a therapist in case of a stress reaction to it.

  1. Lines 408-409: I wonder if more specific information could be added here for what strategies might emerge from the results. It seems like there is already a pretty good sense that the trial will lead to a recurrent of PTSD symptoms.

We have added this sentence:  In addition, this study should pave the way for preventive strategies and specific care to manage or limit traumatic reactivation during legal proceedings such as psycho-education on legal proceedings (with age-related tools: comics, ludic applications, etc.) and the risks of traumatic reactivation,  peers support groups and individual consultations

  1. In terms of limitations, the different connection each participant had with the attack seems like it would have been worth examining. For instance, someone who’s life was in danger might have a different reaction that someone who was indirectly impacted, and it would be valuable to see data on this difference.

The reviewer is right. One of the reviewer suggested that we remove the limitation section as the paper is a protocol. However, we will take the comment into account for future analysis since this information is included in the “14-7” research program.

Round 2

Reviewer 3 Report (New Reviewer)

Comments and Suggestions for Authors

This is the major concern.

No regression performed and no Results section.

Author Response

Dear reviewer, we have chosen to submit an article in the "study protocol" section of healthcare journal. This section is used to present the protocol method without the results. We thank you for your previous comments, which have enabled us to improve the manuscript. Concerning the statistical analysis, we can confirm that "multiple regressions will then be performed to test if sleep measures and somatic symptoms can predict traumatic reactivation symptoms over time". This sentence has been added to the section on statistical analysis.

This manuscript is a resubmission of an earlier submission. The following is a list of the peer review reports and author responses from that submission.

Round 1

Reviewer 1 Report

Comments and Suggestions for Authors

  The title of the article is presented as a protocol. However, pages 6-13 are written as though it will give results of the study. The intention of this article is not clear. Is this an article that explains the reasoning of the protocol of the study was selected….or is this an article that will present findings from this study. In either purpose the article as it stands is largely incomplete. The article is missing reasoning of why the protocol was selected OR is missing the crucial data if it is presenting findings. This article is lacking evidence to support the longitudinal aspect of the study. 

·         This article could provide valuable information once it is in it’s complete form and full intention

The topic of trauma reactivation is quickly gleaned over in the introduction but is a crucial topic of this article. it appears that this is a missing crucial aspect of the article. 

·         Authors should include the country of where Nice is located since this is an international journal

·         Line 36: grammar error, too many “and” use of a comma is more appropriate

·         Line 37: grammar error, “Main analyses comprise variance” – perhaps authors intended to use “will”, “of” or sentence needs to be transitioned to past tense

·         Lines 45-52: written in future tense. Given the current year of 2023 the proper tense is past.

·         Line 46: mentions September 5th to January 15th – it is unclear if these dates cross over a new year mark

·         Line 62-66: authors mention that there are few studies about relapse and intervention but do not explain those that are currently in the literature that are related to this article. This appears to be a large gap in explain what traumatic reactivation is.  

·         Line 73-Line 85: authors appear to be explaining the different ways people could have been exposed to the traumatic event. It appears that authors are alluding to Criteria A of the DSM-5-TR. If so, this is should be included and clarified.

·         Line 88-91: Therapy names are not capitalized but are proper nouns

·         Line 129 is one sentence. Paragraphs are not one sentence.

Comments on the Quality of English Language

A moderate amount of English editing is needed. The tense of the article is written inconsistently (present vs. past tense) and throughout the article there are missing articles/words (of, the, a). 

Reviewer 2 Report

Comments and Suggestions for Authors

This study appears to be aimed at evaluating sleep symptoms and somatic symptoms in a cohort of children who were victims of a terrorist act as the trial commences of the perpetrators. 

The study does not clearly mention how many members of this group are already diagnosed with PTSD, sleep disorders, or other psychiatric diagnosis. Will these be excluded from the study? will there be 2 cohorts, with one containing children with no psychiatric diagnosis and the other with children with a psychiatric diagnosis?

if the study looks at traumatic reactivation only in children with Chronic PTSD, will patients who were never able to adequately obtain relief from sleep symptoms and somatic symptoms of PTSD prior to the trial be excluded from the study?

The protocol spends a lot of words (from lines 95 to 187) talking about topics that are already well-understood. It might be better if they talked about the currently proposed pathophysiological hypothesis about how PTSD leads to sleep symptoms and somatic symptoms. 

Comments on the Quality of English Language

the English language used is readable and appropriate

Reviewer 3 Report

Comments and Suggestions for Authors

Thanks to the authors for their manuscript which I found interesting but which unfortunately lacks experimental parts and data analysis

I suggest some main revisions. 1.  The introduction could be restructured by trying to reduce the number of paragraphs and eliminating paragraphs that are too short. The manuscript begins with a brief description of the attack. A single description of the attack, but brief and concise, could be clearer, adding what is written in paragraph 1.2: - there are several passages to content already exposed. It is good to introduce them only once and in a homogeneous form - points 1.3, 1.4 and 1.5 could be grouped by exposing in summary what is already widely known in the literature. It is in fact known that sleep disturbances represent a symptom of PTSD - Ethical Consideration, Funding, and Registration could be added in paragraph 2 describing the research design - Objectives and hypotheses could be added to paragraph 2. They do not need a separate paragraph. The authors are invited to highlight what their studies add to the scientific panorama. In other words, what does the study add to what is already known in the literature? - The hypotheses appear a bit vague     2.  From line 231 the authors describe the conduct of the study. It is not clear how data collection was controlled There is no clear description of the sample: average age, percentage of males and females. The inclusion and exclusion criteria do not need separate paragraphs which disperse the overall view of the study. The manuscript appears fragmented and repetitive in places   The authors say they excluded participants with low IQ. How was this information collected and what is the average IQ? Normally the procedure is described first and then the tools used   3.    It is preferable to replace Outcomes with Results The results of the correlations, Anova and regressions that the Authors claim to have carried out are missing. The conclusions cannot be verified. It is not clear how the following considerations confirmed the hypothesis The manuscript lacks important and fundamental parts  

Comments on the Quality of English Language the manuscript is understandable

Reviewer 4 Report

Comments and Suggestions for Authors

In the introduction it is stated “After traumatic exposure, sleep disorders are the earliest, most sensitive and persistent symptoms” – is there a citation for this? This seems like an important point to support

The description of the study purpose (Starting with “This research project offers a valuable opportunity”) should be in its own sub-section. Currently it is part of the Somatic Symptoms subsection

What does " Participation in another research with an exclusion period.” In the non-inclusion criteria section mean?

Figure 1 should include a depiction of when the trial occurs relative to the measurements. How long is the trial expected to take? When does the trial start? The Figure includes dates from 2022 – is this a mistake?

Considering this is a protocol/preregistration of a methodology, the statistical analysis section is quite underwhelming and based on the authors’ description of their hypotheses I can envision them conducting many analyses that are not in the current protocol. As an example, the hypothesis states that sleep disturbances before and/or after the beginning of the trial might be associated with PTSD symptoms. How would the authors measure the association between the different stages of sleep measurement and PTSD? What kinds of corrections for multiple comparisons will be used, what types of effect sizes will be reported? As it stands, the authors suggest a very simple data approach that will not really address their hypotheses and leaves open a lot of room for degrees of freedom.

There should be more information as to how the actigraphy/smart watch data will be processed and what data will be used

Comments on the Quality of English Language

“victims ’life” should be “victims’ lives”